# The nature of carotenoid S* state and its role in the nonphotochemical quenching of plants

Davide Accomasso [1,2] ✉, Giacomo Londi [1], Lorenzo Cupellini [1] & Benedetta Mennucci [1] ✉

In plants, light-harvesting complexes serve as antennas to collect and transfer the absorbed energy to reaction centers, but also regulate energy transport by dissipating the excitation energy of chlorophylls. This process, known as nonphotochemical quenching, seems to be activated by conformational changes within the light-harvesting complex, but the quenching mechanisms remain elusive. Recent spectroscopic measurements suggest the carotenoid S* dark state as the quencher of chlorophylls' excitation. By investigating lutein embedded in different conformations of CP29 (a minor antenna in plants) via nonadiabatic excited state dynamics simulations, we reveal that different conformations of the complex differently stabilize the lutein s-trans conformer with respect to the dominant s-cis one. We show that the s-trans conformer presents the spectroscopic signatures of the S* state and rationalize its ability to accept energy from the closest excited chlorophylls, providing thus a relationship between the complex's conformation and the nonphotochemical quenching.

Light-harvesting complexes (LHCs) of oxygenic photosynthetic organisms carry out a twofold function. First, in low-light conditions, LHCs serve as antennas by collecting light and transferring the absorbed energy to the reaction centers, where light photons will be ultimately converted into chemical energy to sustain life[1–3]. Secondly, LHCs act as protectors of the photosystem supercomplexes through a mechanism known as nonphotochemical quenching (NPQ). This mechanism safeguards against photodamage by regulating the energy flow and dissipating excess excitation energy as harmless heat. Consequently, it prevents the initiation of unwanted side reactions, such as the formation of triplet excited states, interactions with molecular oxygen, and the production of reactive oxygen species[4–8].

In higher plants, both the major LHCII and the minor antennas have been proven to participate in the NPQ[9–11]. In particular, the minor CP29 antenna has been identified in earlier works as a pivotal player in the photoprotective function, mediating energy transfer between LHCII and the reaction centers within the photosystem II (PSII)

supercomplex[12–14]. Contrary to the trimeric LHCII, CP29 is a monomeric complex, endowed with 13-14 chlorophylls (Chls)[11,15,16], mostly of type $a$, and 3 carotenoids (Cars), as shown in Fig. 1. Among the Cars, lutein (Lut) is located in the L1 binding site, violaxanthin in the L2 site, and neoxanthin in the N1 site[9,15–17]. More recently, Ruban and co-workers have shown the relevant role of LHCII in the fastest NPQ component (the energy-dependent quenching, qE) in plants that have been deprived of all the minor antennas, thereby questioning the participation of the latter in the quenching[18].

In all cases, the ancillary pigments (the Cars) are thought to play a crucial role in the NPQ, in addition to being essential in protein folding and stability of the LHC and in broadening its energy absorption window[19–21]. Typically, the first singlet excited state ($S_1$) of Cars is dark, i.e., the electronic transition from the ground state ($S_0$) is forbidden, while the upper-lying $S_2$ is the lowest-energy bright state[20–22]. Upon light absorption, $S_2$ decays in a few hundred fs to $S_1$, from which the internal conversion to $S_0$ occurs on ps timescales[22]. This three-state

[1]Department of Chemistry and Industrial Chemistry, University of Pisa, 56124 Pisa, Italy. [2]Present address: Faculty of Chemistry, University of Warsaw, 02-093 Warsaw, Poland. ✉e-mail: davide.accomasso@dcci.unipi.it; benedetta.mennucci@unipi.it

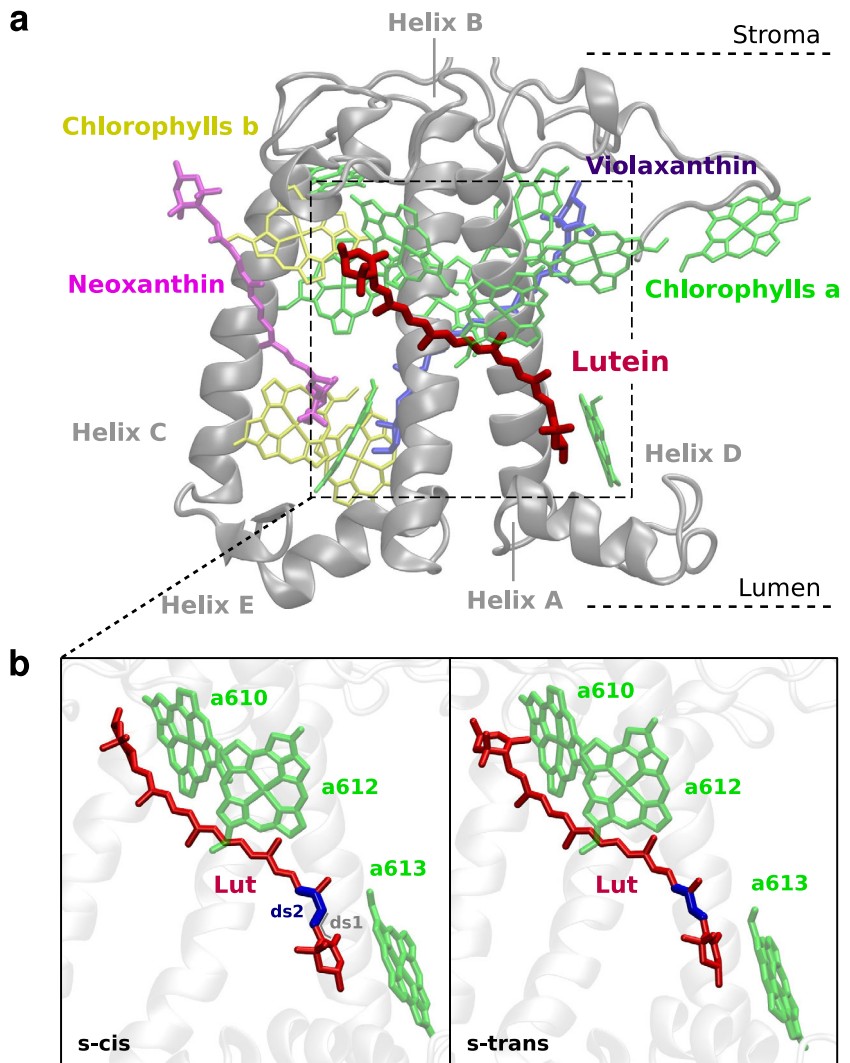

**Fig. 1 | The CP29 light-harvesting complex. a** Side-view of CP29. **b** The L1 site of CP29 for both the lutein (Lut) s-trans and s-cis conformer, including the nearby chlorophylls *a610*, *a612*, and *a613*. The dihedral (ds2), which distinguishes the two conformers, is highlighted in blue. The adjacent dihedral (ds1) is also shown for the s-cis conformer (in gray). The molecular structure of Lut is shown in Supplementary Fig. 1.

picture has been enriched by experimental evidence of additional dark states, whose assignment is still controversial[23–26]. Among these, a so-called S* state has been detected by transient absorption spectroscopy for several Cars in solution or in LHCs[27–30].

Thanks to these unique characteristics of their low-lying electronic excited states, Cars are involved in almost all the different mechanisms proposed over the years in the literature to explain the NPQ[31–34]. In particular, in the qE component of the NPQ, excitation energy transfer (EET) from the lowest excited state of the Chls ($Q_y$) to the neighboring Car(s) is assumed. In this process, Cars can act as quenchers due to their short-lived and optically dark $S_1$ excited state. Another hypothesis is that quenching of the excited Chls goes through a charge transfer from the Cars[32,35–37]. Irrespective of the quenching mechanism, the LHC must be able to switch from its standard light-harvesting function to the quenching one. Nowadays, it is well established that this switch is triggered by a change of pH between the stromal and lumenal side of the embedded thylakoid membrane. The sensing of the pH change by the LHC likely proceeds through an interaction with the PsbS protein[5,7,38]. This interaction can change the

LHC's conformation, thus switching it from a long-lived, light-harvesting state to a short-lived, quenched one.

A thorough investigation of LHCs' photophysics in their natural environment is experimentally challenging. A much better strategy would be that of studying isolated monomeric LHCs. This would allow conditions-controlled experiments and avoid artifacts from phenomena such as singlet-singlet annihilation. Strikingly, isolated LHCs are able to access both light-harvesting and quenched conformations[39–42]. However, in most of the monomeric complexes the light-harvesting conformation is dominant, thus preventing any assessment of the quenching. Using time-resolved fluorescence and transient absorption spectroscopy, Mascoli et al. have observed that a solubilized wild-type monomeric CP29 complex could exist in various emissive states with similar fluorescence spectra but different lifetimes[43]. Most of the complexes have been found in a long-lived state, showing light-harvesting functions, while a smaller but significant fraction of complexes (ca. 16%) are in a quenched state. The authors have been able to assign the spectral features of the quencher to a dark state of Lut, showing photophysical characteristics of the Cars S* state. Similar observations have

been made by Sardar et al. for CP29 inside nanodiscs, which better mimic the near-native thylakoid membrane environment[44].

The existence of the Car S* state with distinct photophysical properties is doubtless, but there are a few interpretations in the literature about its origin[24,27,29,30,45–49]. The Car S* state has been associated both with a hot ground state signal stemming from non-equilibrium vibronic populations[50], with the $S_1$ state in a specific conformation[28,48], or with a twisting of the Car structure while in the excited state[47]. Finally, a combination of these different origins has been proposed to give rise to the S* state features[49,51]. Since the Lut S* state has been revealed as the quencher in CP29, a clear and unequivocal assignment of its features is mandatory in order to investigate the NPQ. This assignment is also crucial to understand the exact mechanism of excited state Chl quenching by Cars.

In this work, we started from the conformational space of CP29 that some of us recently characterized by using enhanced sampling molecular dynamics (MD) techniques[52], and we performed non-adiabatic excited state dynamics simulations of Lut in order to uncover the nature of the S* state and its role in the NPQ. We resorted to a semiempirical quantum mechanics/molecular mechanics (QM/MM) approach and the surface hopping (SH) method. Strikingly, our simulations show that the photophysics of Lut and its spectroscopic properties are strongly tuned by the CP29 conformation through a different stabilization of the minor s-trans conformer with respect to the dominant s-cis conformation (Fig. 1). Moreover, the Lut s-trans conformer shows all the spectroscopic features ascribed to the S* state: (i) a shorter excited state lifetime, and (ii) a blue-shifted excited state absorption peak, when compared to the s-cis conformer. Finally, our calculations indicate that the EET quenching by Lut is more energetically favored for the s-trans conformer. All these findings allow us to elucidate the nature of the quencher S* state in CP29 and to associate it with a specific Lut conformer, whose relevance is controlled by the LHC's conformation.

## Results

The nonadiabatic excited state dynamics simulations presented here are based on the conformational sampling of CP29 generated in a previous work of our group, where an enhanced sampling technique combined with MD simulations was used to largely explore the conformational landscape of the CP29 complex. There, the conformations were divided into six different clusters based on the apoprotein backbone and sidechain conformation (see ref. 52 for more details).

Here, we selected structures from the three main clusters (4, 5 and 6) of ref. 52, as starting coordinates for our simulations. As a result of our selection, we observed two distinct Lut conformers, equally present in cluster 5, namely s-trans and s-cis differing in the first conjugated chain dihedral on the lumenal side (ds2 in Fig. 1). We note in passing that cluster 5 is the most similar to the cryo-EM protein conformation[52]. On the other hand, in clusters 4 and 6 only the Lut s-cis conformer is present with a significant population. This occurs because the different protein conformations stabilize different Car geometries within their binding pockets, as found in ref. 52. From this observation, we decided to group together the selected structures of clusters 4 and 6 to form a first set of conformations, from now on referred to as set A. The structures from cluster 5 were kept in a separate group, named set B in this work.

### The $S_1$ population dynamics and absorption spectra

The excited state SH simulations were initialized mainly in the $S_2$ potential energy surface (PES), according to the transition dipole moment (Supplementary Table 1 and Supplementary Fig. 9). All trajectories rapidly decay through an internal conversion pathway to $S_1$, in a process characterized by a time constant $\tau_2$ of 135 fs for set A and 152 fs for set B. These extracted lifetimes are in good agreement with our recent work on the excited state dynamics of Lut in a methanol solution, where the initial photogenerated population of $S_2$ decays to $S_1$ within 200 fs[53].

Once on the $S_1$ PES, the population decays with a bi-exponential function. In Fig. 2a, we show the $S_1$ population dynamics of the two sets A and B, whereas the time constants extracted from the fitting of the populations are reported in Table 1. Both sets show a faster component ($\tau_1$) of ~1 ps and a slower one ($\tau_1'$) of ~21 ps. Taking into account the relative weights of the bi-exponential function, we computed a weighted average decay time $\tau_1^{avg}$ of 17.1 ps for set A and 13.1 ps for set B. Indeed, in set A, the slower component of the decay has a more prominent weight (ca. 80%) with respect to the faster one, while in set B, the latter gains more weight (41.5%), thereby shifting the overall $S_1$ decay to shorter lifetimes.

In Fig. 2a, we also distinguish the excited state dynamics of the Lut s-trans and s-cis conformers within set B, as they show a very different behavior. If, on the one hand, both the s-trans and s-cis conformers have very similar $S_2$ decay times (Table 1), on the other hand, they show distinct $\tau_1^{avg}$, as $S_1$ in the s-trans conformer is characterized by a shorter lifetime (7.5 ps) than $S_1$ in the s-cis conformer (19.7 ps). As expected,

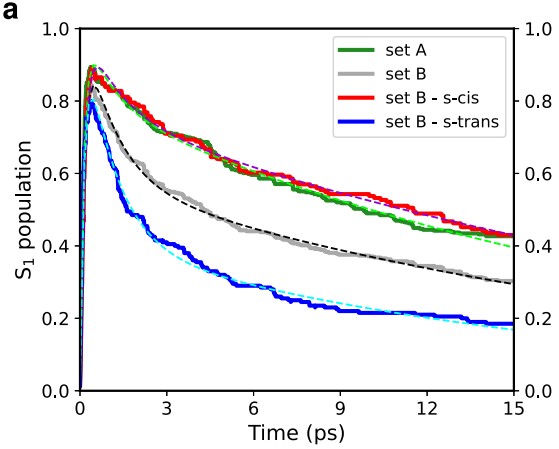
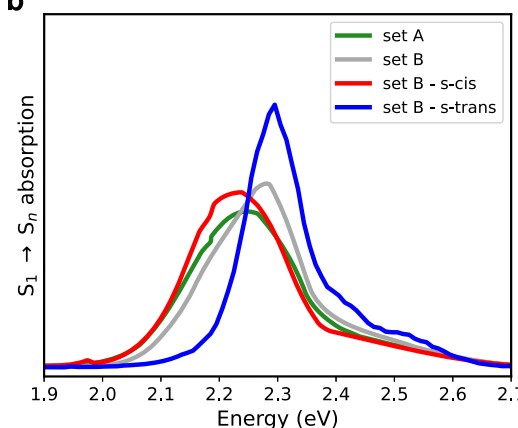

**Fig. 2 | Excited state population dynamics and absorption spectra. a** Populations of the lutein (Lut) $S_1$ state (solid lines) and their fitting functions (dashed lines) obtained in the surface hopping (SH) simulations. The extracted time constants are reported in Table 1. Separate plots of the population dynamics are shown in Supplementary Fig. 4. **b** The $S_1 \rightarrow S_n$ excited state absorption spectra of Lut averaged over all the SH trajectories and the whole simulation time of 15 ps. The distinct contributions from the different $S_n$ states are provided in Supplementary Fig. 5, while the corresponding spectra averaged over time intervals of 1 ps are shown in Supplementary Fig. 6.

**Table 1 | Time constants of excited states decay**

| Set | Time constants | | | |
|---|---|---|---|---|
| | $\tau_2$ | $\tau_1$ | $\tau_1'$ | $\tau_1^{avg}$ |
| A | 135 fs | 1.0 ps (20.1%) | 21.2 ps (79.9%) | 17.1 ps |
| B | 152 fs | 1.1 ps (41.5%) | 21.6 ps (58.5%) | 13.1 ps |
| B - s-cis | 162 fs | 1.2 ps (22.2%) | 25.0 ps (77.8%) | 19.7 ps |
| B - s-trans | 141 fs | 1.1 ps (59.0%) | 16.7 ps (41.0%) | 7.5 ps |

These constants were obtained by fitting the $S_1$ populations of the surface hopping simulations for lutein in CP29: $\tau_2$ (in fs) is the lifetime of $S_2$ and $S_3$, while $\tau_1$ and $\tau_1'$ (in ps) are decay times for $S_1$ (their relative weights and average value are also reported). More details on the fitting procedure are provided in Supplementary Note 1.

the $S_1$ population decay of the Lut s-cis conformer in set B resembles that of set A, which is made of only s-cis conformer, whereas the $S_1$ decay of the s-trans conformer in set B is strikingly different. This suggests that the lumenal side of the Lut has a role in the determination of the $S_1$ decay time.

In the spectroscopic investigations of CP29 two different Car states have been identified[43,44]: (i) the typical $S_1$ state with a lifetime of ~13 ps, and (ii) the quencher S* state characterized by a shorter lifetime of ~6 ps and a blue-shifted excited state absorption (ESA) spectrum compared to $S_1$. In our simulations, the extracted $S_1$ lifetimes for the Lut s-trans and s-cis conformers (7.5 ps and 17–20 ps, respectively) are longer than those observed in spectroscopy, but their ratio (s-trans/s-cis $\simeq$ 0.38–0.44) agrees well with the experiment (S*/$S_1$ $\simeq$ 0.46). Additionally, to facilitate the comparison between our simulations and the spectroscopic measurements, we computed the $S_1 \rightarrow S_n$ ESA spectrum of Lut by performing single-point calculations at each time along the SH trajectories running on the $S_1$ PES. As shown in Fig. 2b, the ESA spectrum of the s-trans conformer peaks at 2.30 eV (539 nm) and is blue-shifted by ~0.06 eV (15 nm) with respect to the ESA spectra of the s-cis conformers in both sets A and B. Therefore, besides showing an intrinsic structural difference induced by the twist along the dihedral angle ds2, the s-trans and s-cis conformers present different excited state lifetimes and ESA spectra. Remarkably, the blue-shift of the Lut s-trans conformer (15 nm) obtained in our simulations is close to the spectroscopic blue-shift of 25–27 nm from the $S_1$ ESA peak (532 nm) to S* (505–510 nm)[43,44]. Based on these results, we can identify the spectroscopic S* state in CP29 with the Lut s-trans conformer, which shows a shorter $S_1$ decay time and a blue-shifted ESA spectrum, compared to the Lut s-cis one. The latter can be instead associated with the spectroscopic $S_1$ state of Lut in CP29[43,44].

Mascoli et al. have observed the transient absorption spectrum of S* to be similar to the spectrum of the Car triplet state ($T_1$), which is formed in the unquenched CP29 complexes[43]. We computed the $T_1 \rightarrow T_n$ and $S_1 \rightarrow S_n$ absorption spectra of Lut using optimized $T_1$ and $S_1$ state structures, respectively (Supplementary Fig. 10). Indeed, the $T_1 \rightarrow T_n$ absorption of the Lut s-cis conformers (set A and B), which we associate with the unquenched conformations of CP29, significantly overlaps with the $S_1 \rightarrow S_n$ spectrum of the Lut s-trans conformer (set B), which in turn we identify with the S* state. These results further support our previous association of the Lut s-trans conformer with the spectroscopic S* state in CP29.

At last, to characterize in more detail the nature of the S* state, we analyzed the electronic character of the $S_1$ state along the SH trajectories of set B. As shown in Supplementary Table 4, the $S_1$ state of the Lut s-trans conformer (which we associate with the S* state) has the characteristics of a typical $S_1/2A_g^-$ Car state, that is a dark state dominated by the HOMO$\rightarrow$LUMO double excitation, with additional large contributions from the HOMO$\rightarrow$LUMO+1 and HOMO-1$\rightarrow$LUMO singly-excited configurations. The same electronic nature is found for the $S_1$ state of the Lut s-cis conformer.

## Comparison between the lutein s-trans and s-cis dynamics

To understand the relationship between the conformational and electronic properties of Lut, we now analyze the structure of the s-trans and s-cis conformers in the ground state ensemble, i.e., at the initial conditions of the excited state dynamics. We focus our analysis on the following geometrical parameters of Lut: (i) the bond-length alternation (BLA); (ii) the two dihedral angles in the lumenal side (ds1 and ds2, Fig. 1); (iii) the distortion around the $\pi$-conjugated C=C bonds. For the latter, we define a $D$ index (Supplementary Equation (4) in Supplementary Note 2) that quantifies the deviation from planarity[54]: a larger distortion is associated to a larger value of $D$. The BLA coordinate is defined as the average difference between single and double bond lengths in the $\pi$-conjugated backbone and variations along this coordinate crucially impact on the Car electronic structure and its excited state properties[53].

Figure 3a shows that in the ground state ensemble the maximum value of the BLA distributions of the s-trans and s-cis conformers is nearly the same, but the s-trans distribution is significantly broader than the s-cis one. Thus, molecular structures in the Lut s-trans conformation explore a larger configurational space already at the ground state. We anticipate that this trend will be kept throughout the SH simulations, both for the ensemble of molecular geometries visited on the $S_1$ state ($S_1$ active) and at the $S_1 \rightarrow S_0$ hops.

It is also interesting to highlight the effect of the dihedral angle ds1, involving the terminal ring of Lut, and the adjacent dihedral angle ds2, defining the s-trans$\rightarrow$s-cis isomerization around the single C-C bond. The distribution of ds1 and ds2 are reported in Fig. 3b, c: while the ds2 distribution reflects a well-defined conformation (at around 180° for the s-trans versus at around 0° for the s-cis), the ds1 distributions of the s-trans and s-cis conformers show peaks at very different values (at around 80° versus at −50°, respectively). Moreover, the ds3 and ds4 dihedral angles, adjacent to ds2, are also significantly affected by the s-isomerization, while the effects on the other C−C dihedral angles (ds5-ds9) vanish (Supplementary Fig. 7). All these findings suggest that the s-isomerization affects the Lut structure mainly at the lumenal side, where the Car is more free to rearrange its conformation[52].

Finally, Fig. 3d shows the total distortion along the C=C bonds. The s-trans geometries are overall more twisted than the s-cis structures, both in the lumenal (Fig. 3e) and the stromal side (Fig. 3f). This indicates that the s-isomerization allows the Car to assume more planar conformations within the protein binding pocket.

As far as regards the structural properties of the Lut s-trans and s-cis conformers during the SH simulations, we characterized both the $S_1$ ensemble and the geometries at which the $S_1 \rightarrow S_0$ transition takes place. Once the $S_1$ state is populated, the BLA coordinate shifts towards smaller values for both conformers (Fig. 4a, b), pointing to a dramatic shortening of the single bonds and an associated elongation of the double bonds compared to the initial BLA distributions (at time = 0). The BLA distributions are centered around zero, meaning that in the $S_1$ state the "single" and "double" bonds have essentially the same length. Notably, the BLA coordinate assumes even smaller values at the $S_1 \rightarrow S_0$ hops. In a previous study on solvated Lut[53], we showed that at negative BLA values the $S_1$ and $S_0$ states are strongly coupled, thus leading to a fast $S_1$ decay. Interestingly, the s-trans conformer presents a significant number of $S_1 \rightarrow S_0$ hopping structures with quite negative BLA values (<−0.02 Å, Fig. 4b, d) and for most of these geometries (~87%, 33 out of 38) the $S_1 \rightarrow S_0$ hop occurs within 2 ps (Fig. 4d). For the s-cis conformer we found similar shifts of the BLA values during the excited state dynamics, but the number of $S_1 \rightarrow S_0$ hops with BLA<−0.02 Å, occurring in the first 2 ps, is much smaller compared to the s-trans conformer (namely, 4 for s-cis versus 33 for s-trans, Fig. 4c, d). In fact, at the $S_1$ state, geometries with negative BLA values are visited to a greater extent for the s-trans conformer, especially in the first 2 ps (Supplementary Fig. 2). We conclude that the larger number of $S_1 \rightarrow S_0$ hops

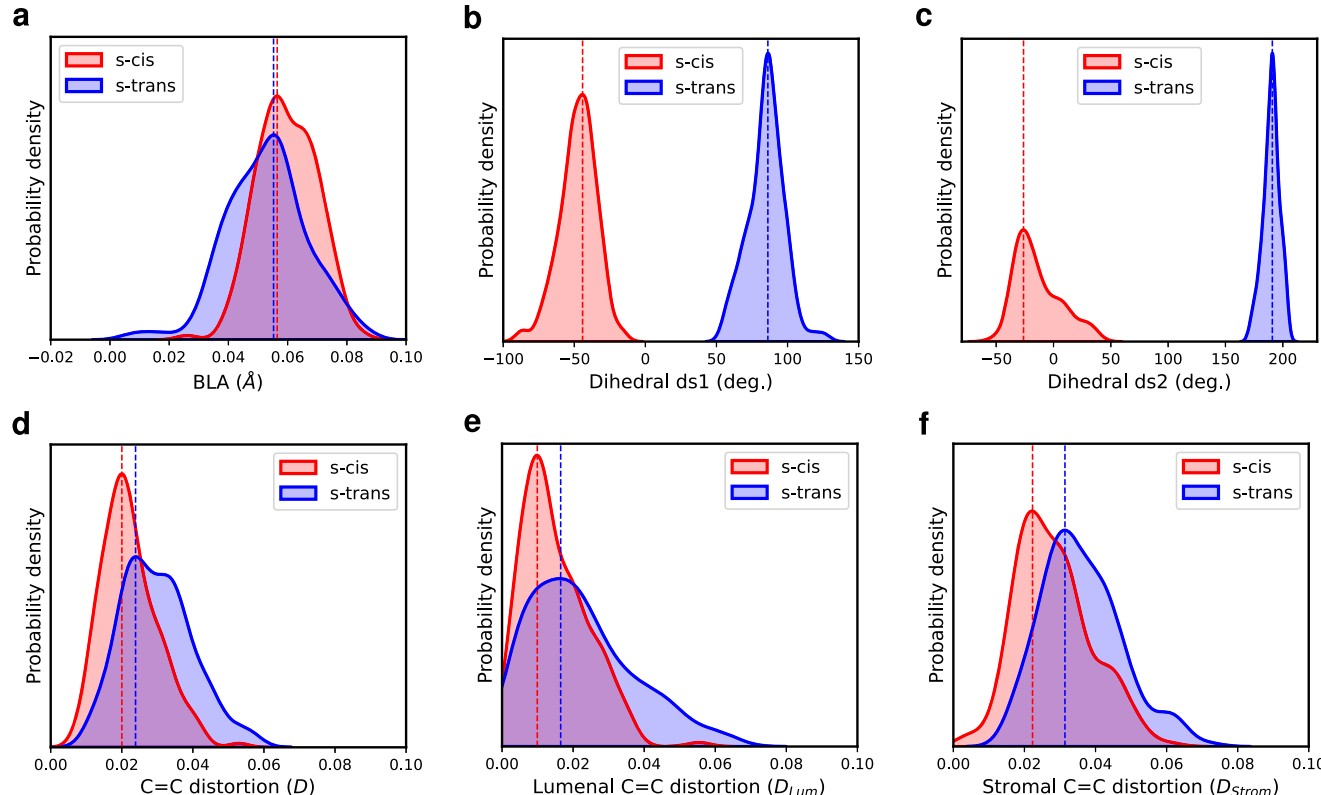

**Fig. 3 | Distributions of different geometrical coordinates of lutein s-trans and s-cis conformers in the ground state ensemble of set B. a** Bond-length alternation (BLA, in Å). **b** Dihedral ds1. **c** Dihedral ds2. **d** Distortion around all the $\pi$-conjugated C=C bonds ($D$ as defined in Supplementary Equation (4) in Supplementary Note 2 for dihedral angles dd1-dd9). **e** Distortion around the C=C bonds in the lumenal side ($D_{Lum}$ for dihedral angles dd1-dd4). **f** Distortion around the C=C bonds in the stromal side ($D_{Strom}$ for dihedral angles dd5-dd9).

with very negative BLA values for the Lut s-trans conformer is the main factor responsible for the larger weight of the fast $\tau_1$ decay component (59% for s-trans versus 22% for s-cis, Table 1).

We further identified another possible $S_1 \rightarrow S_0$ decay channel given by the distortion of the Car $\pi$-conjugated backbone. Figure 5a, b shows the distributions of the distortion around the C=C bonds of the lumenal side ($D_{Lum}$) of the s-trans and s-cis conformers at different times: at time = 0, when $S_1$ is visited ($S_1$ active) and at the $S_1 \rightarrow S_0$ hops. If, at time = 0, Lut in the lumenal side is quite planar (small $D$ values indicate more planar arrangements), as soon as $S_1$ becomes populated, the Car undergoes a significant distortion along its C=C bonds of the lumenal side. This is made possible by the weakening of C=C bonds when Lut is in the $S_1$ state and is consistent with the BLA analysis. By inspecting the contribution of each dihedral angle to $D_{Lum}$ (dd1-dd4), we can observe that, for some of the $S_1 \rightarrow S_0$ hopping geometries with the largest distortion, the first dihedral dd1 shows the largest one, while the other distorted geometries of Lut involve the simultaneous twisting of multiple dihedral angles (Supplementary Fig. 12). Before moving on, we also mention that no photoisomerization event around the C=C bonds was observed during our SH simulations, as shown in Supplementary Fig. 13.

Strikingly, the $S_1 \rightarrow S_0$ hops occur at more distorted geometries with respect to the starting ones (at time = 0) for both the s-trans and s-cis conformers. However, for the s-trans conformer the number of significantly distorted hopping geometries ($D_{Lum} > 0.10$) is larger, especially in the first 2 ps (16 for s-trans versus 7 for s-cis, see the insets in Fig. 5c, d). Most of these $S_1 \rightarrow S_0$ transitions (with $D_{Lum} > 0.10$) are also associated with BLA>−0.02 Å (Supplementary Fig. 3) and, therefore, represent an alternative $S_1$ decay channel, distinct from the one driven by the BLA coordinate. Again, this secondary decay pathway is more

important for the Lut s-trans conformer, thereby speeding up its $S_1$ decay.

**Quenching properties of the lutein s-trans and s-cis conformers**
In the CP29 complex Lut resides in the L1 binding pocket, which is related to the photoprotective function. Lut also interacts more strongly with Chl *a612*, which is in turn associated with a low energy, terminal emitter, excited state of Chls in CP29[9]. In fact, one of the proposed quenching mechanisms is the EET from Chl *a612* to Lut[43]. In addition, there are two more Chls to be taken into account, which are located at the Lut's terminal rings: namely, Chl *a610* at the stromal side and Chl *a613* closer to the lumen.

To investigate the relationship between the different Lut conformations and the quenching mechanism, we computed the electronic couplings between the dark $S_1$ excited state of Lut and the $Q_y$ excitations of the nearby Chls *a610*, *a612*, and *a613* for sets A and B in the ground state ensemble sampled by our simulations. We note that the method used here for calculating the electronic couplings differs from the one used in ref. 52. Here, we used the calculated transition densities localized on each molecular entity, while in the previous study the same densities were approximated in terms of atomic transition charges fitted from the electrostatic potential. As a result, the couplings values here are slightly larger (few tens of cm⁻¹), but still in the same range as the couplings reported in ref. 52. The distributions shown in Fig. 6a−c indicate that the couplings between Lut and the three Chls are quite small in both sets A and B (distribution maxima < 10 cm⁻¹), with minor discrepancies between the two sets. In particular, while the Lut-*a610* and Lut-*a612* couplings are generally larger in set A than those in set B, the opposite is true for the Lut-*a613* coupling. In set B, the computed Lut-Chl couplings are slightly larger for the s-trans conformer than for the s-cis ones. However, the

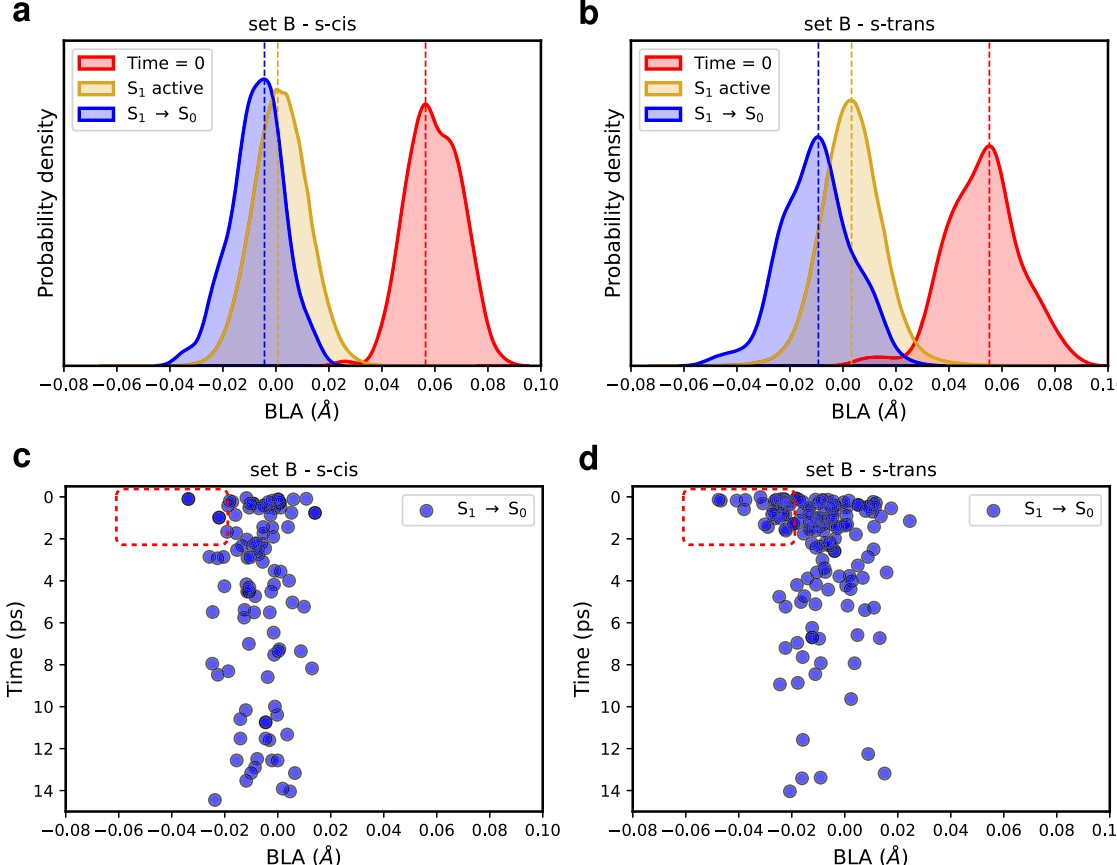

**Fig. 4 | Bond-length alternation. a, b** Distributions of the bond-length alternation (BLA, in Å) values for the starting geometries (at time = 0), those visited in the $S_1$ state ($S_1$ active), and the $S_1 \rightarrow S_0$ hopping geometries, as obtained in the surface hopping simulations of set B. **c, d** BLA as a function of time (in ps) at the $S_1 \rightarrow S_0$ hops. The red insets highlight structures with very negative BLA values (<−0.02 Å) and decay times within 2 ps.

distributions of the Lut-*a610* and Lut-*a612* couplings for the s-trans conformer in set B are very similar to the ones of set A, which includes only the Lut s-cis conformer. This suggests that Lut-Chl couplings are not significantly affected by the change of the Car conformation.

In addition to the Lut-Chl couplings, the EET quenching by Lut is highly sensitive to the energies of the electronic excited states involved. Therefore, we computed the vertical excitation energies of the Lut $S_1$ state and the $Q_y$ state of Chls *a610*, *a612*, and *a613* (Fig. 6d–f). The vertical $Q_y$ excitation energies of Chls lie around 2 eV for all the investigated sets and their distributions are quite narrow, as expected from the rigid molecular structure. The vertical $S_1$ excitation energies, computed at the semiempirical level specifically parametrized for Lut, were additionally benchmarked against DFT/MRCI calculations and the agreement between the two methods was very satisfactory (Supplementary Fig. 11). The $S_1$ distribution of the s-trans conformer in set B, shown in Fig. 6f, is extremely wide, spanning few eV and showing some excitation energies below those of Chls $Q_y$ states. In contrast, this does not occur for the s-cis conformers in either set A or B. This implies that geometrical distortions undergone by Lut in the different conformational sets do really tune the $S_1$ excitation energy. As a result, for some s-trans structures, the energy transfer from Chls to Lut becomes energetically favorable and, in such cases, the Lut s-trans conformer may act as an effective quencher, a role originally ascribed to the S* state[43].

## Discussion

Nonphotochemical quenching, through which LHCs in photosynthetic organisms dissipate the excessive energy in the form of heat, has been a debated topic, and a general consensus has not yet been reached on the quenching mechanism. The most credited hypothesis relies on EET from the excited Chls to Cars, and recent studies on LHCs have identified a peculiar Car electronic state, labeled as S*, as the potentially efficient quencher. Specifically, the S* state has been observed by two independent works for CP29, the minor LHC of plants[43,44].

In this work, we have investigated this hypothesis by performing nonadiabatic SH simulations of Lut in different conformations of CP29. Our simulations show that the excited state lifetime of Lut in CP29 can be tuned by the LHC's conformation through a different stabilization of the Lut conformers. In particular, we identify two significantly different decays of the Lut $S_1$ state, which we associated to the s-trans and s-cis conformers, differing in the first conjugated chain dihedral on the lumenal side. A faster $S_1$ decay (7.5 ps) is obtained in the simulations for the s-trans conformer, while a slower relaxation time (17–20 ps) is found for the s-cis one. The calculations of the $S_1 \rightarrow S_n$ ESA spectra have allowed us to connect our simulations with the reported observations in transient absorption spectroscopy. Specifically, we have assigned the spectroscopic S* state to the $S_1$ state of the s-trans conformer, which, besides a faster-excited state decay, shows also a blue-shifted ESA spectrum, compared to the dominant s-cis conformation. The latter is instead associated with the spectroscopic $S_1$ state[43,44]. By comparing the excited state dynamics of the two conformers, we have attributed the shorter $S_1$ lifetime of the s-trans conformer to the presence of two fast decay channels, almost absent in the s-cis conformer. One of these channels is related to the BLA coordinate, whereas the other is due to the deviation from planarity of the C=C bonds on the lumenal side, where Lut is more free to rearrange its conformation.

Finally, we have assessed the quenching properties of the two conformers. From the calculated electronic couplings between Lut and

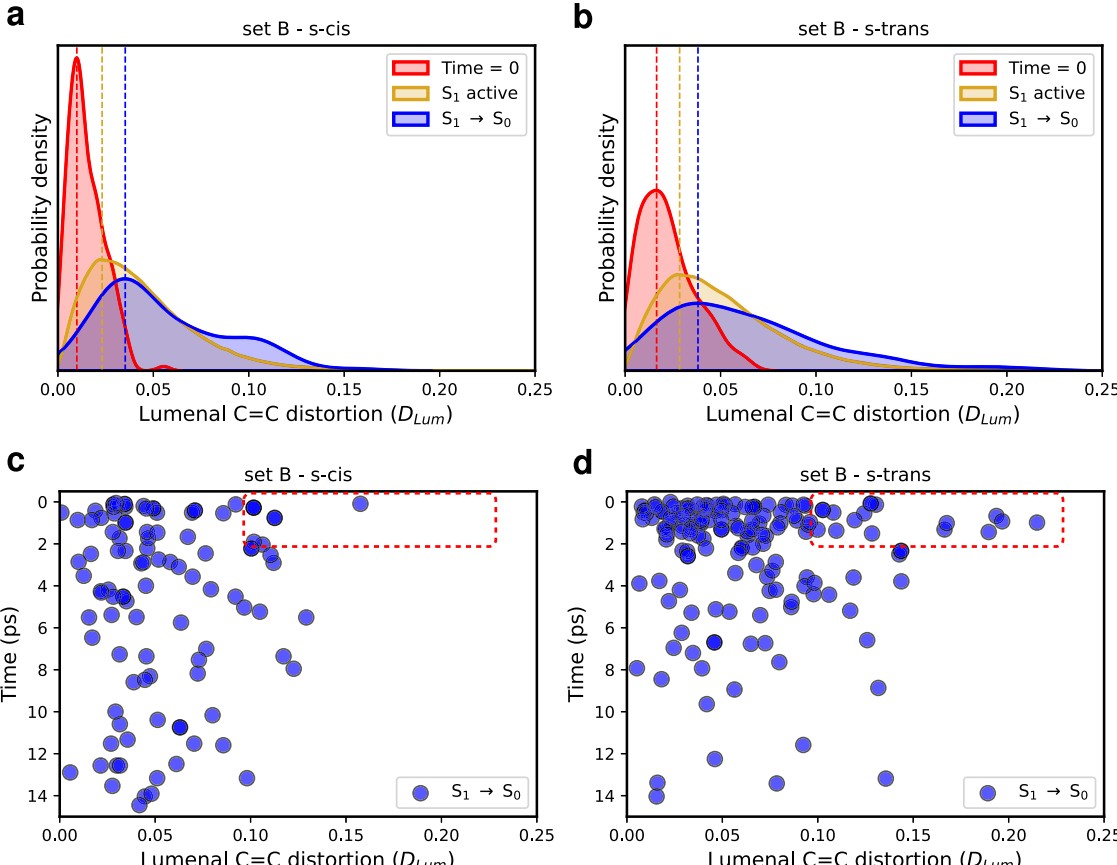

**Fig. 5 | Distortion of the lutein's lumenal side. a, b** Distributions of the distortion around the C=C bonds of the lumenal side ($D_{Lum}$) of lutein (Lut) for the starting geometries (at time = 0), those visited in the $S_1$ state ($S_1$ active), and the $S_1 \rightarrow S_0$ hopping geometries, as obtained in the surface hopping simulations of set B. **c, d** $D_{Lum}$ index as a function of time (in ps) at the $S_1 \rightarrow S_0$ hops. The red insets highlight highly distorted structures with decay times < 2 ps. For the Lut s-trans conformer, the contribution of the individual dihedral angles (dd1-dd4) to $D_{Lum}$ is shown in Supplementary Fig. 12.

the neighboring Chls, we can conclude that the change in Lut conformation does not correspond to a significant change in the Coulomb coupling values, in agreement with a previous study[52]. However, the Lut s-trans conformer presents a large distribution of vertical $S_1$ energies spreading below the $Q_y$ state of the nearby Chls. Conversely, this never occurs for the s-cis conformer. This suggests that EET (and quenching) from excited Chls to Lut is favored when Lut is in the s-trans conformation. Therefore, in addition to associating the $S_1$ state of Lut s-trans with the spectroscopic S* state, our simulations also show that the latter can act as a quencher of Chl excitation in CP29.

## Methods

### Nonadiabatic surface hopping simulations

We resorted to a hybrid QM/MM scheme with electrostatic embedding on a subsystem of the CP29 complex. The Car Lut was treated at the QM level by using the semiempirical configuration interaction method based on floating occupation molecular orbitals (FOMO-CI)[55,56], with the AM1 form of the semiempirical Hamiltonian[57] and the optimized parameters of Lut reported in our previous work[53]. For these calculations, an active space of 6 electrons in 9 $\pi$-type MOs was used, with a Gaussian energy width for floating occupation of 0.1 Hartree, and a determinant space including all single and double excitations within the orbital active space. Lut was embedded in a solvated model membrane, as investigated in ref. 52, where the MM protein environment (within 25 Å from Lut) was described using the parameters of the AMBER ff14SB[58] and lipid14[59] force fields for protein and lipids, respectively, and ad-hoc parameters for the pigments (the Chls and Cars)[60,61]. In particular, the parameters for the Cars were specifically

fitted to describe structural effects on the excitation properties[61]. Furthermore, in the MD simulations, only the MM residues within 18 Å of Lut were allowed to move, while all the other MM atoms were kept frozen. A more detailed account is reported in Supplementary Table 3.

The nonadiabatic excited state dynamics of Lut in CP29 was simulated using the "fewest switches" SH method[62], by employing a locally diabatic representation for the time evolution of the electronic wave function[56] and the overlap decoherence correction scheme[63]. A total of 782 SH trajectories, namely 396 for set A and 386 for set B of CP29 (Supplementary Table 1), were propagated for 15 ps, with starting conditions (the initial nuclear coordinates and velocities, and the starting electronic state) sampled from QM/MM ground state thermal equilibrations at 300 K. These simulations were performed using a development version of the MOPAC2002 code[64], interfaced with the TINKER 6.3 package[65], in which the SH method and the QM/MM semiempirical FOMO-CI technique were implemented. More details on the SH simulations and the thermal equilibrations are provided in Supplementary Methods 1 and 2, respectively. Trajectories and data analysis were performed with in-house Fortran 90 and Python scripts.

### Electronic couplings calculations

The Lut-Chl electronic couplings were determined by evaluating the Coulomb integrals between the transition densities of each chromophore, computed in separate QM/MM calculations[66]. For Chls *a610*, *a612*, and *a613*, the ground and $Q_y$ states computed using the semiempirical FOMO-CI method with the AM1 original parameters[57], an active space of 6 electrons in 6 $\pi$-type MOs, a Gaussian energy

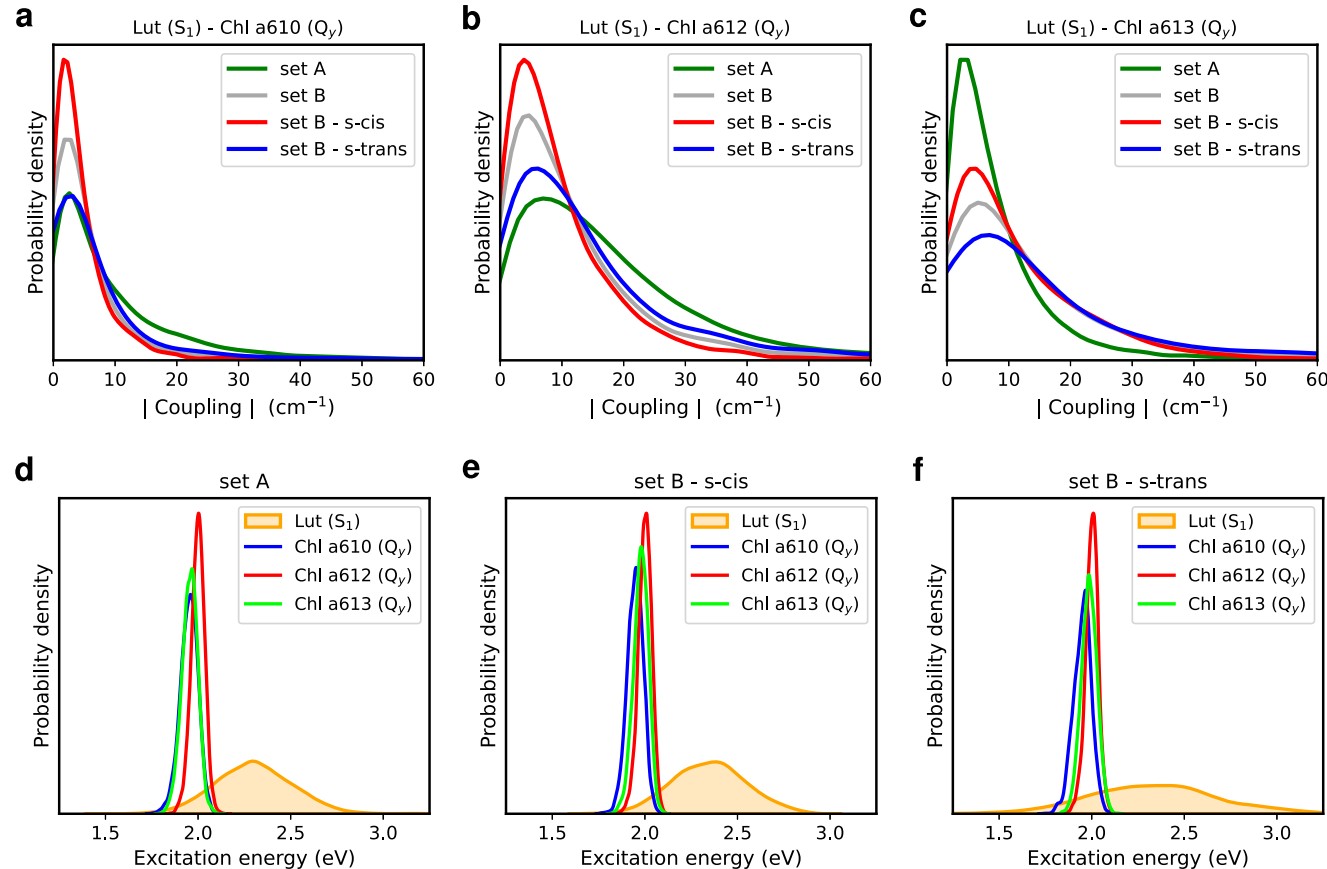

**Fig. 6 | Electronic couplings and vertical excitation energies. a–c** Distributions of the electronic Coulomb couplings (in cm$^{-1}$) between the lutein (Lut) S$_1$ state and the Q$_y$ states of chlorophylls (Chls) *a610*, *a612*, and *a613*. **d–f** Distributions of the vertical excitation energies (in eV) for Lut S$_1$ and Chl Q$_y$ states. The reported couplings and excitation energies were computed along the QM/MM ground state thermal equilibrations of Lut in CP29 (time interval from 0.5 ps to 2.0 ps).

width for floating occupation of 0.1 Hartree, and a determinant space comprising all possible excitations within the orbital active space.

### Benchmark of the lutein S$_1$ energy
To compare with results obtained by FOMO-CI calculations, the vertical S$_1$ excitation energies of Lut were also computed with the DFT/MRCI technique[67]. In running such calculations, we selected a set of structures (20 each for Lut s-trans and s-cis in set B) sampled along the QM/MM thermal equilibrations. The DFT calculations were performed with the BHLYP exchange-correlation functional, the def2-SVP basis set, and by using the point charge embedding scheme for the MM part surrounding the Lut. For the MRCI step, we took advantage of the latest released R2022 Hamiltonian, which improves double excitation energies[68]. The DFT/MRCI calculations were carried out with the TURBOMOLE 7.5 software[69,70].

### Reporting summary
Further information on research design is available in the Nature Portfolio Reporting Summary linked to this article.

## Data availability
Data supporting all the findings of this study are available within the article and its supplementary file. All data, including those referring to Figs. 2–6 and those shown in the Supplementary Figures and Supplementary Tables are available in the Zenodo repository: https://zenodo.org/records/10221745.

## Code availability
The code used in this work is available from the corresponding authors upon request.

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

## Acknowledgements

All authors acknowledge funding by the European Research Council under the grant ERC-AdG 786714 (LIFETimeS). L.C. and B.M. also acknowledge financial support from ICSC-Centro Nazionale di Ricerca in High-Performance Computing, Big Data, and Quantum Computing, funded by the European Union-NextGenerationEU-PNRR, Missione 4 Componente 2 Investimento 1.4. All authors thank Giovanni Granucci and Maurizio Persico of the University of Pisa for sharing the development version of the MOPAC2002 code interfaced with the TINKER 6.3 package.

## Author contributions

B.M. acquired funding; D.A. performed the ground state thermal equilibrations; D.A. and G.L. performed the surface hopping and QM/MM simulations; G.L. performed the DFT/MRCI calculations; D.A. and G.L. carried out the data analysis; D.A., L.C., and B.M designed the research; all authors wrote and edited the paper and approved its final version.

## Competing interests

The authors declare no competing interests.
