## [Peer Review File · Nature Communications]

The nature of carotenoid S* state and its role in the nonphotochemical quenching of plantsREVIEWER COMMENTS

Reviewer #1 (Remarks to the Author):

This is a very well-written account of a computational project that investigates the nature of the "S*" carotenoid state, a dark state suggested to act as chlorophyll excitation quencher. The study focuses on the lutein of the light harvesting protein CP29. The results demonstrate that the embedded lutein can be stabilized by the protein conformation in two conformers, s-cis and s-trans. It is shown that the excited state of the latter decays more slowly. Specific factors related to bond-length-alternation and out-of-plane distortion are identified that affect uniquely the s-trans conformer. Crucial spectroscopic properties are computed, which support assignment of the S1 state of s-trans lutein to the "S*" spectroscopic state (excited-state absorption spectra), but also suggest that excitation energy transfer can occur from chlorophyll to s-trans lutein, explaining the quenching role. Overall this is a very convincing work that can have direct impact on the study of the specific system but also more generally in the field of simulation of pigment-protein complexes. The methodology is suited to the problem and is explained with sufficient detail. I have only a few minor comments and questions: 1) Is it possible to employ a polarizable embedding approach with the computational protocol used here? 2) Freezing a portion of the MM can result in problems during MD such as instability due to unphysical force redistribution; have the authors witnessed any issues with the stability of their trajectories or have they taken any special measures? 3) I would really like to see a proper discussion of the nature of the S* state (as promised in the title) in terms of electronic structure. For now, the manuscript equates the S* to the S1 carotenoid state, but there is no analysis at the quantum chemical level of the actual nature of the S1 and S2 states. It is not even clear that the method employed is able to provide a reasonable description of these states, particularly S1. Perhaps this is literature information (if not, relevant discussion and perhaps supporting calculations should be presented), but it is in any case necessary to have for a paper in such journal.

Reviewer #2 (Remarks to the Author):

The study by Accomasso et al. reports on the nature of the carotenoid S* state within the monomeric CP29 light harvesting complex of higher plants, by employing simulations at atomic scale. Enhanced sampling of the CP29 configurational space has been achieved in a previous publication of the group (Nat Commun 2021, 12: 7152). The authors extend the analysis of the simulations reported therein, and additionally perform excited state non-adiabatic dynamics (surface hopping, SH) on a subset of the configurational space identified for CP29. The study confirms the findings/ proposals of two older studies (Liguori et al. Nat Commun. 2017, 8, 1994; Mascoli et al., Chem 2019, 5:2900–2912) on the twist of the Lutein carotenoid end-ring that acts as the switch between the light harvesting and quenching states of the light harvesting complexes.

Noteworthy findings include the interpretation of the different life-times of S1 and S* states, the work adds significant insight into the field of non-photochemical quenching and it can be worth publishing in Nat Commun. Enough details are provided in the methods for reproducibility. However, there are some points that needed further discussion, for solid conclusions on the association between the S* state and the s-trans conformation of the Lutein within CP29. In detail:

(1) The authors state that "In higher plants, both the major (LHCII) and the minor antennas, in particular CP29, participate in NPQ, with CP29 having a pivotal role in the photoprotective

function". However, in a recent publication (Journal of Experimental Botany 2020, 71, 12:3626–3637), it is reported that the major component of NPQ (qE) is present in plants that lack all minor antennas. This indicates that given an enhanced transthylakoid membrane ΔpH , only the major LHCII contributes to qE. The authors might want to rephrase their statement, regarding the key role of minor antennas in NPQ.

(2) The authors have assigned the 539nm peak to the s-trans (S^*) state of Lutein, and a ~554nm peak (15nm shift) to the s-cis ($S1$) carotenoid conformation. It might be easier to make the connection to the experiments, by introducing also a wavelength axis in Fig. 2b. S-cis excited state lifetime is calculated at ~20ps and that of s-trans (S^*) at 7.5ps. The experimental evidence suggests a 532 nm ($S1$) / 505 nm (S^*) assignment, with the S^* spectrum to be similar to the carotenoid triplet state spectrum. Could the authors provide a carotenoid triplet spectrum based on the s-trans population at a similar level of theory? A match between triplet and S^* spectra on the same s-trans population could be a further proof of the S^* to s-trans assignment.

(3) In Figs. d-f there is a somewhat marginal difference between excitation energy distributions between s-cis and s-trans conformations. Could the authors provide an estimation of the errors/ accuracy of the method employed, so that the marginal difference becomes significant?

(4) The authors should discuss the trigger for the carotenoid twist between s-cis and s-trans conformations. How exactly is this transition correlated with conformational changes of the CP29 protein scaffold? How do changes in the pH at the luminal side of CP29 affect (and why) the population of the s-trans/ s-cis conformations? All these can be directly associated with the transition from light harvesting to the quenched state.

Reviewer #3 (Remarks to the Author):

Accomasso et al studied the molecular mechanism behind the nonphotochemical quenching in the complex CP29 which is important to prevent photodamage in plants. In this computational study the authors suggest that a conformational change in the lutein (a carotenoid found in CP29) to be responsible for the spectroscopically measured S^* state signature.

This work presents results from a combination of quantum mechanics and molecular mechanics simulations. In particular, nonadiabatic molecular dynamics simulations of the lutein in CP29. Further, electronic coupling and excitation energy calculations are presented for lutein and three neighboring chlorophylls.

This computational study is done on a high level, considering the full environment and long nonadiabatic dynamics simulations using a semi-empirical method. There are a few major concerns that should be addressed before the manuscript can be considered for publication:

i) The entire mechanism is based on the presence of the s-trans lutein. Can the authors exclude the possibility that the s-trans isomer formation of lutein is an artifact of force field simulations?

I'm raising this concern because molecules with conjugated pi-electrons are impossible to simulate with a classical force field because the pi-conjugation is an electronic effect which is not available in a classical force field. If a one bond in a conjugated system changes it will affect several bonds of the lutein, which is not captured by classical force fields.

Moreover, in a conjugated molecule such as lutein, the rotation of a single bond should have a high barrier. Similar to the smallest conjugated molecule butadiene, the single bond from the dihedral ds_2 in lutein, should have a partial double bond character.

What are the exact dihedral parameters used for lutein? The references citing the force field parameters in this manuscript refer to other publications which makes it difficult to understand.

ii) Similar to the previous point, the dihedral ds_1 has a value of close to 90 degrees in the ground state (Fig.3b), which significantly breaks the pi-conjugation. If such a geometry is optimized at an ab initio level of theory I expect it to planarize. My impression is that this is an artifact of the force field MD simulation. Although the authors run a QM/MM simulation in the ground state, the duration of 1.5 ps might not be sufficient to relax the highly distorted geometry from classical simulations.

The highly distorted s-trans geometry of lutein (ds_1 & ds_2) would also explain the faster S1 relaxation times compared to the more planar s-cis geometry found in this study. A broken conjugation would also explain the blue shift in ESA which was found for s-trans geometries and the wider distribution of the excitation energies in lutein.

iii) It is not clear why the manuscript starts with the description of nonadiabatic dynamics of lutein. In the naturally occurring process this would be the second step after excitation energy transfer. It would make sense to start from the chlorophyll excitation calculation, including the computation of electronic coupling and comparison of the range of excitation energies of lutein (Fig.6 e-f). Subsequently, the nonadiabatic trajectories of lutein could be analyzed based on the excitation energy of lutein and its overlap with chlorophyll Qy state energy.

What if only a subset of s-cis and s-trans trajectories is associated with higher quenching? This is an important question because also the excited state lifetimes have a distribution. So maybe those trajectories which hop on a short timescale also have a better overlap of the excitation energy with chlorophyll a?

In addition, the author could explain why they have excluded S2 in lutein as a possible source for NPQ. The geometries of lutein are distorted therefore it can be expected that S1 and S2 are mixing.

iv) The analysis of the lutein geometry after the surface hop is completely missing. Is there some isomerization? If there is any new product formed, it could give a hint to experimentalists to validate the proposed NPQ mechanism. This information is important because it is possible to analyze lutein using HPLC and check for different conformers (including s-trans).

v) A comparison of computed and measured lifetimes and also ESA shifts is missing in the manuscript.

vi) BLA is a good descriptor for the bond length alternation because all the single and double bonds are coupled by the shared pi-conjugated system. In contrast, there is no coupling between the dihedrals. Therefore, the parameter D (which describes the collective distortion of the conjugated chain) is not suitable to analyze the distortion of lutein (Fig.5). Also it would not allow to distinguish cases where one double bond is highly twisted or several dihedrals

are slightly twisted, because D is calculated from the average. It would be much better to analyze each dihedral or at least find the dihedral which has the largest distortion. This would help to understand those geometries highlighted in the red box of Fig.5c,d.

Reviewer 1

This is a very well-written account of a computational project that investigates the nature of the "S*" carotenoid state, a dark state suggested to act as chlorophyll excitation quencher. The study focuses on the lutein of the light harvesting protein CP29. The results demonstrate that the embedded lutein can be stabilized by the protein conformation in two conformers, s-cis and s-trans. It is shown that the excited state of the latter decays more slowly. Specific factors related to bond-length-alternation and out-of-plane distortion are identified that affect uniquely the s-trans conformer. Crucial spectroscopic properties are computed, which support assignment of the S1 state of s-trans lutein to the "S*" spectroscopic state (excited-state absorption spectra), but also suggest that excitation energy transfer can occur from chlorophyll to s-trans lutein, explaining the quenching role. Overall this is a very convincing work that can have direct impact on the study of the specific system but also more generally in the field of simulation of pigment-protein complexes. The methodology is suited to the problem and is explained with sufficient detail. I have only a few minor comments and questions:

Authors' Reply: We thank the Reviewer for the positive comments.

(1) Is it possible to employ a polarizable embedding approach with the computational protocol used here?

Authors' Reply: In principle it is certainly possible but here we have used QM/MM calculations in the electrostatic embedding scheme, as a polarizable embedding approach has not yet been implemented for surface hopping nonadiabatic dynamics. The development and implementation of a polarizable embedding approach for the computational protocol used in this study (using for example the AMOEBA force field) will be the subject of future work.

(2) Freezing a portion of the MM can result in problems during MD such as instability due to unphysical force redistribution; have the authors witnessed any issues with the stability of their trajectories or have they taken any special measures?

Authors' Reply: We thank the Reviewer for this question. However, along our QM/MM trajectories we did not observe any instability issue. In fact, we employed a quite large cutoff for freezing atoms, namely 18 Å from the QM Lut. This choice allowed us to include in the MM moving subsystem about 10000 atoms, which is more than half of the whole MM part (see the new **Table S3** added in the Supporting Information). This is enough to ensure that kinetic energy can be redistributed between the system and environment within the time scale of our QM/MM simulations. On the other hand,

freezing the outer shell allows us to avoid evaporation of external waters or small molecules.

(3) I would really like to see a proper discussion of the nature of the S^* state (as promised in the title) in terms of electronic structure. For now, the manuscript equates the S^* to the S_1 carotenoid state, but there is no analysis at the quantum chemical level of the actual nature of the S_1 and S_2 states. It is not even clear that the method employed is able to provide a reasonable description of these states, particularly S_1 . Perhaps this is literature information (if not, relevant discussion and perhaps supporting calculations should be presented), but it is in any case necessary to have for a paper in such journal.

Authors' Reply: We thank the Reviewer for pointing out that in the original manuscript the nature of S^* was not clear in terms of electronic structure. We have indeed analyzed the nature of the S_1 state along the SH trajectories for Lut s-trans and s-cis in set B. We now report the weights of the most important electronic configurations in the new **Table S4** (added in the Supporting Information). As it can be seen, the S_1 state of Lut s-trans (which we associate with S^*) has the characteristics of a typical $S_1/2A_g^-$ carotenoid dark state, dominated by the HOMO \rightarrow LUMO double excitation, with additional large contributions from the HOMO \rightarrow LUMO+1 and HOMO-1 \rightarrow LUMO singly-excited configurations. The same electronic configurations, with similar weights, dominate the S_1 state of Lut s-cis (Table S4). This means that the electronic structure/character of S^* is akin to that of "regular" S_1 .

We have also added a paragraph at the end of Section 2.1:

"At last, to characterize in more detail the nature of the S^* state, we analyzed the electronic character of the S_1 state along the SH trajectories for Lut s-trans in set B. As it can be seen in Table S4, the S_1 state of Lut s-trans (which we associate with S^*) has the characteristics of a typical $S_1/2A_g^-$ carotenoid state, that is a dark state dominated by the HOMO \rightarrow LUMO double excitation, with additional large contributions from the HOMO \rightarrow LUMO+1 and HOMO-1 \rightarrow LUMO singly-excited configurations. The same electronic nature is found for the S_1 state of s-cis Lut."

Reviewer 2

The study by Accomasso et al. reports on the nature of the carotenoid S* state within the monomeric CP29 light harvesting complex of higher plants, by employing simulations at atomic scale. Enhanced sampling of the CP29 configurational space has been achieved in a previous publication of the group (Nat Commun 2021, 12: 7152). The authors extend the analysis of the simulations reported therein, and additionally perform excited state non-adiabatic dynamics (surface hopping, SH) on a subset of the configurational space identified for CP29. The study confirms the findings/ proposals of two older studies (Liguori et al. Nat Commun. 2017, 8, 1994; Mascoli et al., Chem 2019, 5:2900–2912) on the twist of the Lutein carotenoid end-ring that acts as the switch between the light harvesting and quenching states of the light harvesting complexes. Noteworthy findings include the interpretation of the different lifetimes of S1 and S* states, the work adds significant insight into the field of non-photochemical quenching and it can be worth publishing in Nat Commun. Enough details are provided in the methods for reproducibility. However, there are some points that needed further discussion, for solid conclusions on the association between the S* state and the s-trans conformation of the Lutein within CP29. In detail:

Authors' Reply: We thank the Reviewer for the positive comments.

(1) The authors state that "In higher plants, both the major (LHCII) and the minor antennas, in particular CP29, participate in NPQ, with CP29 having a pivotal role in the photoprotective function". However, in a recent publication (Journal of Experimental Botany 2020, 71, 12:3626–3637), it is reported that the major component of NPQ (qE) is present in plants that lack all minor antennas. This indicates that given an enhanced transthylakoid membrane ΔpH , only the major LHCII contributes to qE. The authors might want to rephrase their statement, regarding the key role of minor antennas in NPQ.

Authors' Reply: We thank the Reviewer for pointing out this recent publication. We acknowledge that the role of CP29 in NPQ is not completely established. We emphasize, however, that the ability of CP29 to perform quenching *in vitro* (in both detergent and nanodiscs) is anyways indisputable. Nonetheless, there is still evidence that CP29 participates in NPQ *in vivo*, as CP29 mutants show different NPQ properties (see ref. 11). The evidence that *in vivo* CP29 participates in NPQ cannot be truly neglected, but taking into account the main message of the contribution by Ruban and coworkers, we have rephrased our sentence as:

"In higher plants, both the major (LHCII) and the minor antennas have been proven to participate in NPQ. In particular, the minor CP29 antenna has been identified in earlier works as a pivotal player in the photoprotective function, mediating energy transfer between the major LHCII antenna complexes and the reaction centers within the

photosystem II (PSII) supercomplex. [...] More recently Ruban and coworkers have shown the relevant role of the LHCII complex in the fastest NPQ component (the energy-dependent quenching, qE) in plants that have been deprived of all the minor antennas, thereby questioning the participation of the latter in the quenching.“

(2) The authors have assigned the 539nm peak to the s-trans (S*) state of Lutein, and a ~554nm peak (15nm shift) to the s-cis (S1) carotenoid conformation. It might be easier to make the connection to the experiments, by introducing also a wavelength axis in Fig. 2b. S-cis excited state lifetime is calculated at ~20ps and that of s-trans (S*) at 7.5ps. The experimental evidence suggests a 532 nm (S1) / 505 nm (S*) assignment, with the S* spectrum to be similar to the carotenoid triplet state spectrum. Could the authors provide a carotenoid triplet spectrum based on the s-trans population at a similar level of theory? A match between triplet and S* spectra on the same s-trans population could be a further proof of the S* to s-trans assignment.

Authors' Reply: We thank the Reviewer for this suggestion. Indeed, one of the characteristics of the S* feature in the work by Mascoli *et al.* was the similarity with the triplet spectrum. To compare the triplet excited-state absorption with the singlet (S1) one, we computed the $T_1 \rightarrow T_n$ and $S_1 \rightarrow S_n$ absorption spectra of Lut for various configurations using structures optimized for the T1 and S1 state, respectively (see the new **Figure S10** added in the Supporting Information). We found that the $T_1 \rightarrow T_n$ absorption of Lut s-cis (set A and B), which we associate with the unquenched conformations of CP29, significantly overlaps with the $S_1 \rightarrow S_n$ spectrum of Lut s-trans (set B), which in turn we identify with S*. Therefore, these results further support the association of Lut s-trans with the spectroscopic S* state.

Accordingly, we have added a new paragraph in Section 2.1, which is as follows:

“Mascoli *et al.* have observed the transient absorption spectrum of S* to be similar to the spectrum of the Car triplet state (T1), which is formed in the unquenched CP29 complexes.⁴³ We computed the $T_1 \rightarrow T_n$ and $S_1 \rightarrow S_n$ absorption spectra of Lut using optimized T1 and S1 state structures, respectively (see Figure S10). Indeed, the $T_1 \rightarrow T_n$ absorption of Lut s-cis (set A and B), which we associate with the unquenched conformations of CP29, significantly overlaps with the $S_1 \rightarrow S_n$ spectrum of Lut s-trans (set B), which in turn we identify with S*. These results further support our previous association of Lut s-trans with the spectroscopic S* state in CP29.”

(3) In Figs. d-f there is a somewhat marginal difference between excitation energy distributions between s-cis and s-trans conformations. Could the authors provide an estimation of the errors/ accuracy of the method employed, so that the marginal difference becomes significant?

Authors' Reply: To give a more complete answer to this important question we have now employed a more accurate QM method as benchmark for the semiempirical R-AM1/FOMO-CI method employed in our simulations. We computed the S_1 vertical excitation energy of Lut using the DFT/MRCI technique for a set of structures sampled along the QM/MM ground state thermal equilibrations. In the new **Figure S11** (added in the Supporting Information) we have compared the DFT/MRCI (reference) energies with those computed with the semiempirical R-AM1/FOMO-CI. The two methods are in very good agreement with each other, with a mean absolute deviation smaller than 0.1 eV for both the s-trans and s-cis conformers of Lut in set B. Moreover, the DFT/MRCI calculations confirm that the distribution of S_1 energies for Lut s-trans is broader than the corresponding one for Lut s-cis (see **Figure S11, panel c**).

We have also added a sentence in Section 2.3:

“The vertical S_1 excitation energies, computed at the semiempirical level specifically parametrized for Lut, were additionally benchmarked against DFT/MRCI calculations and the agreement between the two methods was very satisfactory (see Figure S11).”

(4) The authors should discuss the trigger for the carotenoid twist between s-cis and s-trans conformations. How exactly is this transition correlated with conformational changes of the CP29 protein scaffold? How do changes in the pH at the luminal side of CP29 affect (and why) the population of the s-trans/ s-cis conformations? All these can be directly associated with the transition from light harvesting to the quenched state.

Authors' Reply: We thank the Reviewer for this question. To the best of our knowledge, changes in the luminal pH should not directly affect the conformational balance in CP29. It is instead the PsbS protein that senses luminal pH and induces a conformational change in CP29. However, *isolated* CP29 can reversibly access various conformations, as evidenced by the time-resolved fluorescence and transient absorption measurements of Mascoli *et al.* (*ref. 43*). They demonstrated that a good fraction (16%) of CP29 in solution is in a quenched conformation. These results were confirmed by Sardar *et al.* (*ref. 44*) in membrane nanodiscs.

In a previous work by some of us (*ref. 52*), it was demonstrated that the CP29 apoprotein has access to multiple luminal conformations. Such conformations were identified by a clustering method accounting *only* for the protein conformation, which still showed a significant difference in the dihedral conformation of the Lut. We can deduce that the protein conformation is coupled to the population of the s-trans/s-cis conformations of Lut. In this work, “Set A” and “Set B” are extracted from different basins in the conformational landscape of CP29, and for this reason have different populations of s-trans/s-cis Lut conformers.

Reviewer 3

Accomasso et al studied the molecular mechanism behind the nonphotochemical quenching in the complex CP29 which is important to prevent photodamage in plants. In this computational study the authors suggest that a conformational change in the lutein (a carotenoid found in CP29) to be responsible for the spectroscopically measured S^* state signature. This work presents results from a combination of quantum mechanics and molecular mechanics simulations. In particular, nonadiabatic molecular dynamics simulations of the lutein in CP29. Further, electronic coupling and excitation energy calculations are presented for lutein and three neighboring chlorophylls. This computational study is done on a high level, considering the full environment and long nonadiabatic dynamics simulations using a semi-empirical method. There are a few major concerns that should be addressed before the manuscript can be considered for publication:

Authors' Reply: We thank the Reviewer for the comments.

(1) The entire mechanism is based on the presence of the s-trans lutein. Can the authors exclude the possibility that the s-trans isomer formation of lutein is an artifact of force field simulations? I'm raising this concern because molecules with conjugated pi-electrons are impossible to simulate with a classical force field because the pi-conjugation is an electronic effect which is not available in a classical force field. If a one bond in a conjugated system changes it will affect several bonds of the lutein, which is not captured by classical force fields. Moreover, in a conjugated molecule such as lutein, the rotation of a single bond should have a high barrier. Like the smallest conjugated molecule butadiene, the single bond from the dihedral ϕ_2 in lutein, should have a partial double bond character. What are the exact dihedral parameters used for lutein? The references citing the force field parameters in this manuscript refer to other publications which makes it difficult to understand.

Authors' Reply: We understand the Reviewer's concerns, and we agree that classical force fields may struggle to describe the conjugation effects in these molecules. . However it has to be stressed that the force field used here is not a "general" force field, but it was specifically fitted to obtain accurate electronic properties of lutein (see ref. 61).

The rotation around a single bond does have a quite high barrier, but this barrier is not so high for single bonds close to the rings. For the rotation around ϕ_2 , DFT calculations at the B3LYP/6-31G(d) and M06-2X/6-31G(d) levels give barriers of around 9.5 and 8 kcal/mol, respectively, whereas the barrier predicted at the force field level is around 7.5 kcal/mol. Thus, the force field does a reasonable job at capturing the barrier for rotation around this single bond.

(2) Similar to the previous point, the dihedral ds1 has a value of close to 90 degrees in the ground state (Fig.3b), which significantly breaks the pi-conjugation. If such a geometry is optimized at an ab initio level of theory I expect it to planarize. My impression is that this is an artifact of the force field MD simulation. Although the authors run a QM/MM simulation in the ground state, the duration of 1.5 ps might not be sufficient to relax the highly distorted geometry from classical simulations. The highly distorted s-trans geometry of lutein (ds1 & ds2) would also explain the faster S1 relaxation times compared to the more planar s-cis geometry found in this study. A broken conjugation would also explain the blue shift in ESA which was found for s-trans geometries and the wider distribution of the excitation energies in lutein.

Authors' Reply: Given the presence of a terminal cyclohexenyl ring, the dihedral ds1 cannot be planar, as shown by DFT ground state optimization (ds1 = -47° for Lut optimized at B3LYP/6-31G(d) level in gas-phase, see below). For Lut s-trans, MD simulations indicate that the twisting around ds1 is more pronounced in CP29 (ds \cong 90°), compared to the gas-phase. We have previously verified that the MD structures of Lut in CP29 and DFT-optimized ones present similar dihedral angles (supplementary figure 11 of ref. 52). We take this as additional evidence that the MM force field does not introduce a significant bias in the distortion of the dihedral angles of lutein.

(3) It is not clear why the manuscript starts with the description of nonadiabatic dynamics of lutein. In the naturally occurring process this would be the second step after excitation energy transfer. It would make sense to start from the chlorophyll excitation calculation, including the computation of electronic coupling and comparison of the range of excitation energies of lutein (Fig.6 e-f). Subsequently, the nonadiabatic trajectories of lutein could be analyzed based on the excitation energy of lutein and its overlap with chlorophyll Qy state energy. What if only a subset of s-cis and s-trans trajectories is associated with higher quenching? This is an important question because also the excited state lifetimes have a distribution. So maybe those trajectories which hop on a short timescale also have a better overlap of the excitation energy with chlorophyll a? In addition, the author could explain why they have excluded S2 in lutein as a possible source for NPQ. The geometries of lutein are distorted therefore it can be expected that S1 and S2 are mixing.

Authors' Reply: To give a clearer picture of the Lutein's excited state dynamics, we decided to first analyze the Lut alone, focusing on the differences between conformations, and then on understanding the quenching of Chl excitation.

We understand the Reviewer's suggestion of correlating the hop time with the (S_1) excitation energy. However, such an analysis would not be sensible in the context of stochastic surface hopping simulations. In these simulations, we represent a *single nuclear wavepacket* with a swarm of trajectories with different initial conditions extracted from a ground state ensemble. Moreover, the hopping time is determined by a stochastic algorithm (*i.e.*, not uniquely determined by the initial conditions). Therefore, it is not possible to analyze or give a meaning to a cherry-picked single trajectory, but one always needs to analyze an ensemble. What we can do is separate *a priori* the trajectories based on the initial conditions: it is clear that s-trans and s-cis structures (as well as structures from sets A and B) belong to completely uncorrelated nuclear wavepackets. The same reasoning applies to the classical-ensemble excitation energies of Figure 6.

Finally, we would like to remark that, although we report in Figure 6 the couplings and excitation energies involving S_1 , any mixing between S_1 and S_2 is already taken into account within our calculations. In fact, S_1 is (most of the time) the dark state, but it mixes with the bright state, acquiring some dipole strength. This determines the variation in coupling values seen in Figure 6a-c. Conversely, S_2 itself is too high in energy to contribute to the quenching (experimentally, the S_2 absorption starts at ~540 nm, whereas Chl emission is around 680 nm).

(4) The analysis of the lutein geometry after the surface hop is completely missing. Is there some isomerization? If there is any new product formed, it could give a hint to experimentalists to validate the proposed NPQ mechanism. This information is important because it is possible to analyze lutein using HPLC and check for different conformers (including s-trans).

Authors' Reply: We thank the Reviewer for this question. However, in our SH simulations in CP29 we did not observe any photoisomerization of C=C double bonds of Lut (see the distributions of dihedrals dd1-dd9 of Lut at the beginning, 0 ps, and at the end, 15 ps, of the SH trajectories, reported in the new **Figure S13**).

In Section 2.2 we have added:

“Before moving on, we also mention that no photoisomerization event around the C=C bonds was observed during our SH simulations, as shown in Figure S13.”

(5) A comparison of computed and measured lifetimes and also ESA shifts is missing in the manuscript.

Authors' Reply: We thank the Reviewer for this remark.

To address this point, we have added the missing information in Section 2.1 in which we explicitly compare the spectroscopic lifetimes and ESA shift with the corresponding quantities obtained in our simulations. The new text reads as follows (see the fourth paragraph in Section 2.1):

“In the spectroscopic investigations of CP29 two different Car states have been identified:^{43, 44} (i) the typical S_1 state with a lifetime of ~ 13 ps, and (ii) the S^* quencher state characterized by a shorter lifetime of ~ 6 ps and a blue-shifted ESA spectrum compared to S_1 . In our simulations, the extracted S_1 lifetimes for the Lut s-cis and s-trans conformers (i.e., 17-20 ps and 7.5 ps, respectively) are longer than the spectroscopic times by $\sim 30\%$, but their ratio (s-cis/s-trans $\cong 2.3$ -2.7) agrees well with that of the spectroscopic lifetimes ($S_1/S^* \cong 2.2$).”

and

“Remarkably, the ESA blue-shift of Lut s-trans of 15 nm obtained in our simulations is close to the spectroscopic blue-shift of 25-27 nm from the S_1 ESA peak (532 nm) to S^* (505-510 nm).^{43, 44}”

(6) BLA is a good descriptor for the bond length alternation because all the single and double bonds are coupled by the shared pi-conjugated system. In contrast, there is no coupling between the dihedrals. Therefore, the parameter D (which describes the collective distortion of the conjugated chain) is not suitable to analyze the distortion of lutein (Fig.5). Also it would not allow to distinguish cases where one double bond is highly twisted or several dihedrals are slightly twisted, because D is calculated from the average. It would be much better to analyze each dihedral or at least find the dihedral which has the largest distortion. This would help to understand those geometries highlighted in the red box of Fig.5c,d.

Authors' Reply: We have analyzed the contribution of each C-C=C-C dihedral angle to the collective distortion of Lut in the Luminal side (D_{Lum}). In **Figure S12** we reported such contributions for the $S_1 \rightarrow S_0$ hopping geometries with the largest distortion. It can be observed that, while in some cases dihedral dd1 shows the largest distortion, several distorted geometries of Lut involve the simultaneous twisting of multiple dihedrals. The distortion index D_{Lum} we used is able to capture both situations within a single descriptor. We also note that a similar descriptor was used by some of us to capture the distortion of the canthaxanthin in the orange carotenoid protein (new ref 54).

We have added a sentence in Section 2.2:

“By inspecting the contribution of each C-C=C-C dihedral angle to the collective distortion index D_{Lum} , we can observe that, for some of the $S_1 \rightarrow S_0$ hopping geometries with the largest distortion, the first dihedral $dd1$ shows the largest one, while the other distorted geometries of Lut involve the simultaneous twisting of multiple dihedral angles (Figure S12).”

REVIEWERS' COMMENTS

Reviewer #1 (Remarks to the Author):

The authors have fully addressed the comments on the initially submitted version and revised the manuscript appropriately. I have no further comments on this version of the manuscript.

Reviewer #2 (Remarks to the Author):

The authors have adequately amended the manuscript, based on the reviewer comments. Thus, I recommend publication in Nat Comm.

Reviewer #3 (Remarks to the Author):

The response of the authors has addressed all the points raised during the revision. This work is suitable for publication.